# The response of wildfire regimes to Last Glacial Maximum carbon dioxide and climate

Olivia Haas[1,2], Iain Colin Prentice[1,2], Sandy P. Harrison[1,3]

[1]Leverhulme Centre for Wildfires, Environment and Society, Imperial College London, South Kensington, London SW7 2BW, UK

[2]Georgina Mace Centre for the Living Planet, Department of Life Sciences, Imperial College London, Silwood Park Campus, Buckhurst Road, Ascot SL5 7PY, UK

[3]Geography & Environmental Science, University of Reading, Whiteknights, Reading RG6 6AH, UK

*Correspondence to*: Olivia Haas (o.haas20@imperial.ac.uk)

**Abstract**

Climate and fuel availability jointly control the incidence of wildfires. The effects of atmospheric $CO_2$ on plant growth influence fuel availability independently of climate; but the relative importance of each in driving large-scale changes in wildfire regimes cannot easily be quantified from observations alone. Here, we use previously developed empirical models to simulate the global spatial pattern of burnt area, fire size and fire intensity for modern and Last Glacial Maximum (LGM; ~ 21,000 ka) conditions using both realistic changes in climate and $CO_2$ and sensitivity experiments to separate their effects. Three different LGM scenarios are used to represent the range of modelled LGM climates. We show large, modelled reductions in burnt area at the LGM compared to the recent period, consistent with the sedimentary charcoal record. This reduction was predominantly driven by the effect of low $CO_2$ on vegetation productivity. The amplitude of the reduction under low $CO_2$ conditions was similar regardless of the LGM climate scenario and was not observed in any LGM scenario when only climate effects were considered, with one LGM climate scenario showing increased burning under these conditions. Fire intensity showed a similar sensitivity to $CO_2$ across different climates but was also sensitive to changes in vapour pressure deficit (VPD). Modelled fire size was reduced under LGM $CO_2$ in many regions but increased under LGM climates because of changes in wind strength, dryness (DD) and diurnal temperature range (DTR). This increase was offset under the coldest LGM climate in the northern latitudes because of a large reduction in VPD. These results emphasise the fact that the relative magnitudes of changes in different climate variables influence the wildfire regime and that different aspects of climate change can have opposing effects. The importance of $CO_2$ effects imply that future projections of wildfire must take rising $CO_2$ into account.

## 1. Introduction

Climate influences the occurrence of wildfires both through fire weather, which affects the probability of wildfire start and spread, and the long-term establishment of vegetation which is strongly controlled by temperature and precipitation (Bradstock, 2010; Pausas & Ribeiro, 2013). It has been suggested that current climate change, driven by increasing atmospheric $CO_2$ levels, will increase wildfire risk in many regions through increased fuel dryness whilst potentially reducing wildfire risk in some regions due to decreasing fuel availability (e.g. Abatzoglou et al., 2019a; Bowman et al., 2020; Harrison et al., 2021; Rogers et al., 2020). However,

atmospheric $CO_2$ levels also affect fuel loads independently of climate through physiological effects on photosynthesis which cascade into plant growth rates (Bond et al., 2003; Bond & Midgley, 2012; Kgope et al., 2010). Much emphasis has been placed on recent and future changes in fire weather (see e.g. Abatzoglou et al., 2019; Betts et al., 2015; Flannigan et al., 2013; Jolly et al., 2015). However, increases in atmospheric $CO_2$ concentrations promote vegetation productivity, thus altering fuel availability and loads, as well as affecting fuel types through e.g. woody thickening (Buitenwerf et al., 2012; Donohue et al., 2013; Knorr et al., 2016; Martin Calvo et al., 2014; Martin Calvo & Prentice, 2015; Pausas, 2015). Fuel properties have different effects on different aspects of the fire regime, with fire size strongly constrained by fuel continuity and fire intensity limited by fuel loads (Archibald et al., 2013; Haas et al., 2022). Thus, $CO_2$-induced changes in vegetation properties will most likely affect these aspects of wildfire regimes differently.

One reason the impact of $CO_2$ on wildfires is poorly constrained is the difficulty of isolating it based on observations alone. Satellite records only span ~25 years, a relatively short period to monitor the effect of changing $CO_2$ levels on the vegetation properties that influence wildfires. Furthermore, changes in atmospheric $CO_2$ levels and climate are temporally correlated, and since both affect vegetation, it difficult to attribute changes in observations to one or the other. An alternative approach is to use process-based fire-enabled vegetation models which explicitly account for the physiological effects of $CO_2$ and can be used to examine the temporal and spatial patterns of wildfires under different conditions. Process-based models have been used to examine the impact of climate and atmospheric $CO_2$ changes on both vegetation and wildfire at the last glacial maximum (LGM; 21,000 years ago) (Martin Calvo et al., 2014; Martin Calvo & Prentice, 2015). The LGM is a useful out-of-sample experiment since the climate forcing is of similar magnitude as the change expected by the end of the century in high-end scenarios, though of opposite sign (Kageyama et al., 2021). The LGM had a generally colder and drier climate than today, with $CO_2$ levels ~ 185 ppm. Palaeorecords show reduced vegetation productivity and forest cover (Harrison & Prentice, 2003; Kaplan et al., 2016; Moreno et al., 2018), and ice core and sedimentary charcoal records indicate reduced biomass burning globally (Albani et al., 2018; Harrison et al., 2022; Marlon et al., 2016; Rubino et al., 2016). Although this reduction could reflect the colder and drier conditions, model experiments suggests that low $CO_2$ also played a crucial role. Experiments using the coupled biogeography and biogeochemistry model BIOME4 (Kaplan et al., 2003) showed that it was necessary to include the direct effect of $CO_2$ to simulate observed global and regional reduction in forest cover during the glacial (Bragg et al., 2013; Harrison & Prentice, 2003). Similarly, Martin Calvo et al. (2015) showed that low $CO_2$ was necessary to simulate the observed reduction of biomass burning in LGM experiments using the LPX fire-enabled vegetation model.

In this analysis, we use three empirical models (Haas et al., 2022) to explore the relative importance of climate and of $CO_2$ on the global spatial patterns of burnt area, fire size and fire intensity. We performed two experiments under realistic modern $CO_2$ and climate conditions (MOD climate/MOD $CO_2$ and LGM climate/LGM $CO_2$). We also performed two counterfactual sensitivity experiments to quantify the sensitivity of each wildfire property to climate and $CO_2$ independently (MOD climate/LGM $CO_2$ and LGM climate/MOD $CO_2$). Comparisons to LGM charcoal records from the Reading Palaeofire Database (RPD) (Harrison et al., 2022) were used to examine which experiments provided the most realistic spatial patterns.

**2. Methods**

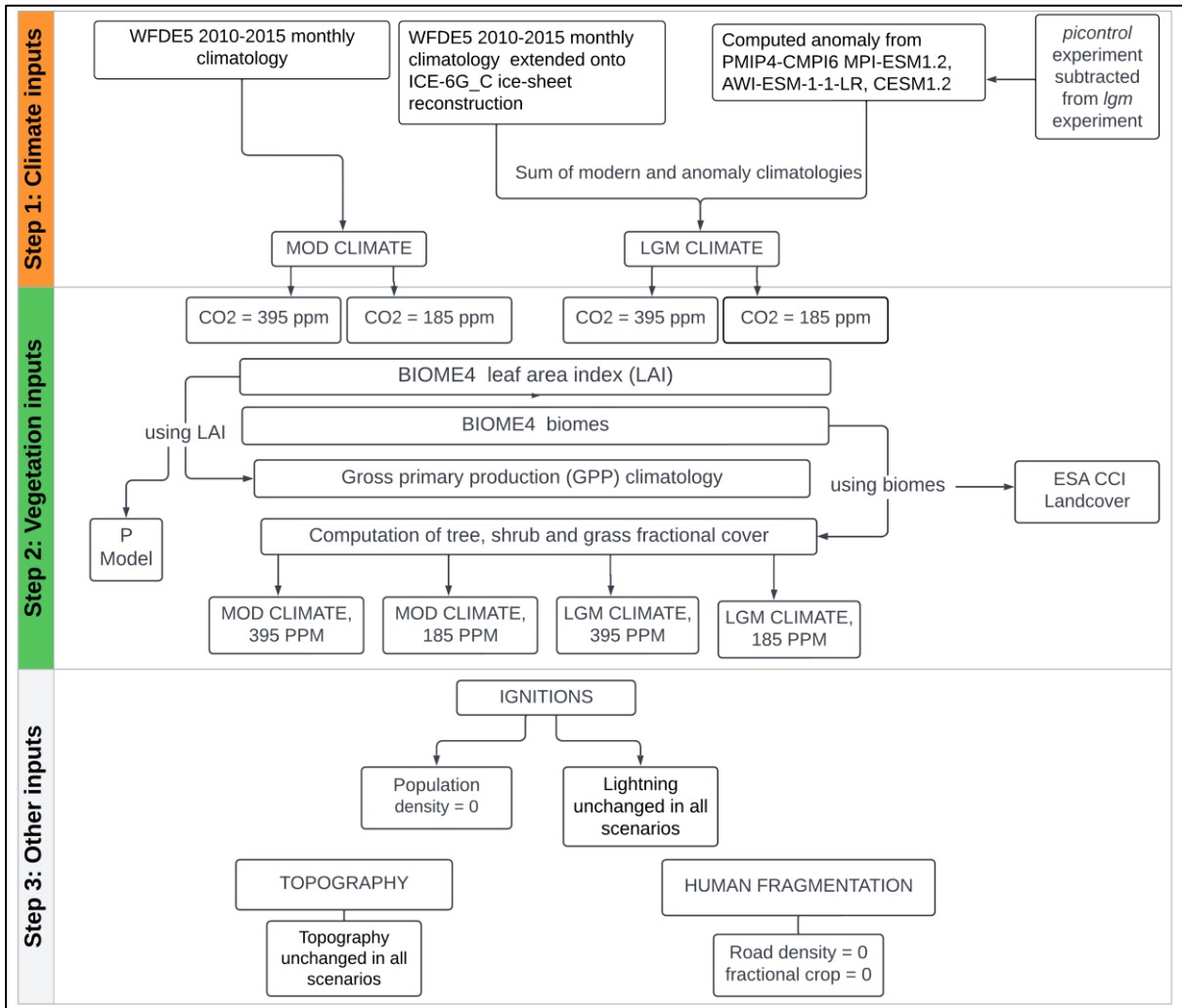

**Figure 1.** Flowchart of the method to obtain each of the four scenarios: MOD climate and MOD $CO_2$, LGM climate and LGM $CO_2$, MOD climate and LGM $CO_2$ and LGM climate and LGM $CO_2$.

Haas et al (2022) developed empirical models of the global spatial patterns of burnt area (BA), fire size (FS) and fire intensity (FI) using generalised linear modelling (GLM) of modern observations. Here we use these models to simulate the global spatial patterns of burnt area (BA), fire size (FS) and fire intensity (FI) under four climate/ $CO_2$ scenarios (Figure 1). We used two realistic scenarios: (a) MOD climate and $CO_2$ conditions and (b) LGM climate and $CO_2$. We ran two sensitivity experiments (a) combining MOD climate and LGM $CO_2$ and (b) combining LGM climate and MOD $CO_2$ levels. The empirical models use climate, vegetation, topography, lightning ignitions, land cover, road density and human population density as predictors to represent the environmental controls on each of the wildfire properties.

Modern (MOD) climate data (daily temperature (T), daily precipitation (P), photosynthetic photon flux density (PPFD), monthly wind speeds (wind), vapour pressure deficit (VPD), monthly specific humidity (huss), cloud cover (cld), monthly pressure (Pa)) were obtained from the WFDE5 bias-adjusted ERA5 database (Cucchi et al., 2020) for 2010 to 2015. The number of monthly dry days (DD) (days with ≤ 1mm of precipitation), monthly diurnal temperature range (DTR) (daily maximum temperature – daily minimum temperature) and monthly

vapour pressure deficit (VPD), a function of specific humidity, temperature and pressure were all calculated
following the methodology in Haas et al. (2022). Seasonal climatologies were derived for all variables eliminating
inter-annual variability. For each grid cell, values from the month with (on average) the maximum number of DD,
the largest DTR, and the highest VPD were selected. Wind speed value was taken from the hottest month of the
year (determined from the WFDE5 2 m air temperature (Cucchi et al., 2020)). For lightning, the mean value over
the seasonal climatology was selected. A seasonality predictor to account for wet vs dry seasons was constructed
by dividing the range of monthly values from the seasonal DD climatology by the mean value of all 12 months.
Expanded ice sheets in North America, Fennoscandia, Greenland, and Antarctica resulted in global sea levels ~
120 m lower than today at the LGM. The modern climate data were extrapolated out onto the exposed shelves
using the ICE-6G_C (Peltier et al., 2015) boundary conditions and a nearest neighbour approach from the
*GeoInterpolation* package in R.

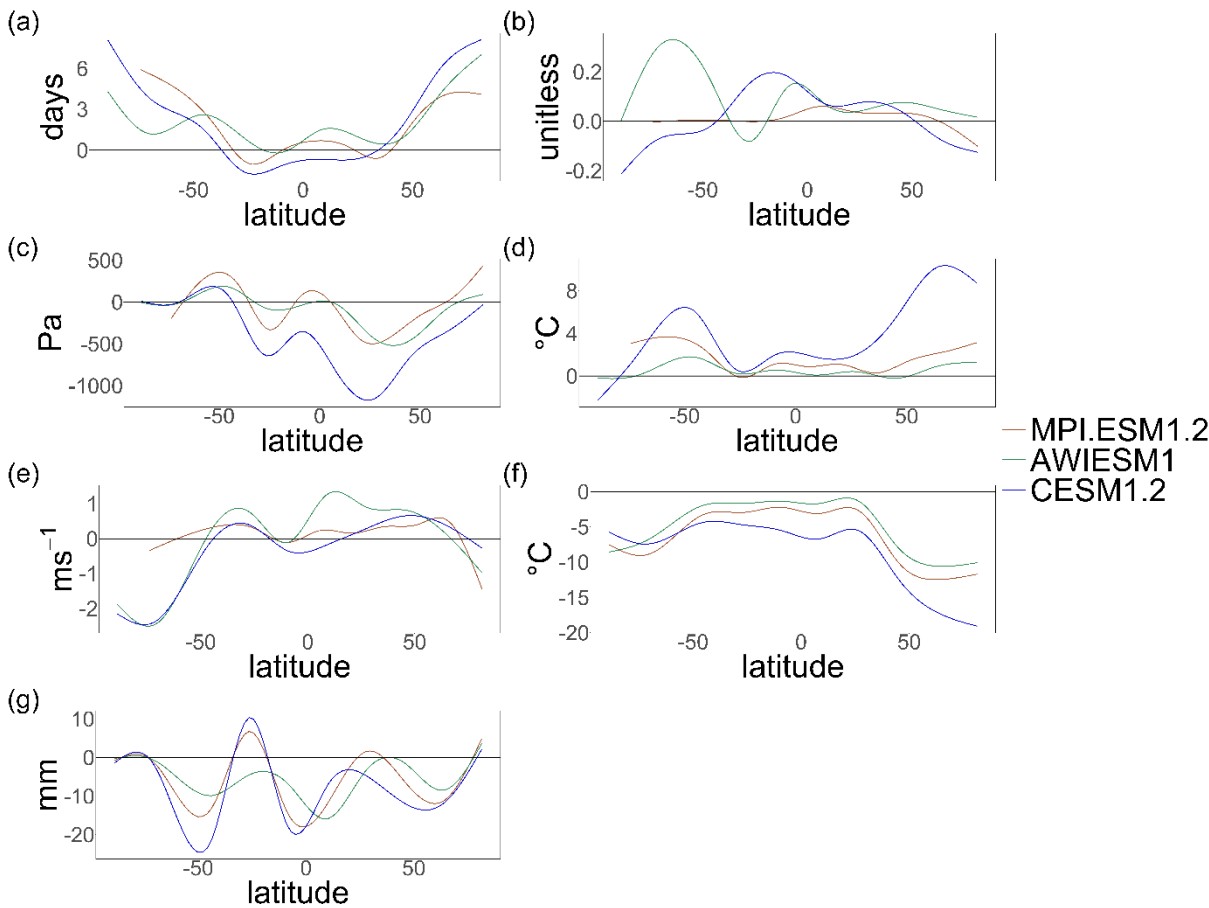


**Figure 2.** Latitudinal distribution of the LGM-MOD climate anomalies for MPI.ESM1.2 (orange), AWI-
ESM1.2 (pink) and CESM1.2 (brown) for (a) the maximum number of dry days, (b) the seasonality of dry days,
(c) maximum monthly VPD, (d) maximum monthly DTR, (e) maximum monthly mean wind speeds, (f) mean
monthly temperature and (g) mean monthly total precipitation. The zero-intercept line represents no change
between LGM and MOD climate, with negative values representing lower values at the LGM and positive
values representing higher values at the LGM.

LGM climate data were obtained from three models participating in the Palaeoclimate Modelling Intercomparison Project (PMIP) contribution to the sixth phase of the Coupled Model Intercomparison Project (CMIP6), AWI-ESM-1-1-LR (short name: AWIESM1) (Lohmann et al., 2020; Sidorenko et al., 2015), MPI_ESM1.2 (Mauritsen et al., 2019), CESM1.2 (F. Li et al., 2013; Tierney et al., 2020) to represent a range of LGM climates (Figure 2). A seasonal climatology was derived for each climate variable from the PMIP *picontrol* experiment (pre-industrial conditions, PI) and the PMIP *lgm* experiment of the PMIP4-CMIP6 simulations. The difference between the PI and LGM values (LGM-PI climate anomalies) were calculated and added to the MOD climatology (LGM-MOD climate anomalies) (see Figure 1). We use the term climate anomalies to refer to the difference between the MOD climatology for each climate variable and the computed bias-adjusted LGM climatology for the same variable, consistent with the PMIP4 protocol (Kageyama et al., 2017). The use of anomalies is designed to minimise the impact of systematic model biases on the derived climate. This approach provided three LGM climate scenarios, resulting in twelve experiments for BA, FS and FI respectively.

We obtained MOD and LGM vegetation and gross primary production (GPP) using the coupled biogeography and biogeochemistry model BIOME4 (Kaplan et al., 2003) and a simple optimality-based model of GPP, the P Model (Wang et al., 2017; Stocker et al., 2020). BIOME4 was used to simulate biome distribution with modern day climate data (T, P, cld) setting $CO_2$ levels to 395 ppm (the annual mean from 2010-2015) and 185 ppm in turn. LGM biome distributions were simulated using the three different LGM scenarios, again setting $CO_2$ levels to 395 ppm and 185 ppm respectively. We derived mean fractional tree, shrub, and grass cover for each of these twelve experiments using the mean values for each biome from ESA CCI Landcover (W. Li et al., 2018). We also calculated fAPAR for each experiment from the leaf area index (LAI) computed by BIOME4 and obtained fractional cover of $C_4$ plants (see S1). We computed global monthly $C_3$ and $C_4$ photosynthesis using the P model using appropriate combinations of climate (T, VPD, ppfd and Pa), BIOME4-derived fAPAR and $CO_2$ concentration for the MOD and LGM scenarios (see Figure 1). Total GPP was calculated as:

$$GPP_{monthly} = GPP_{c3}(1 - C4_{fraction}) + GPP_{c4}C4_{fraction} , \qquad (1)$$

with $GPP_{c3}$ and $GPP_{c4}$ representing monthly $C_3$ and $C_4$ GPP values from the P Model and $C4_{fraction}$ representing the fractional C4 cover from BIOME4 (see Table1).

| Scenario | Modern climate | MPI_ESM1.2 | AWIESM1 | CESM1.2 LGM |
|---|---|---|---|---|
| Modern CO₂ (395 ppm) | 149.37 | 106.63 | 112.06 | 88.44 |
| LGM CO₂ (185 ppm) | 66.54 | 55.49 | 69.61 | 50.37 |

**Table 1.** Total annual gross primary production (GPP) (in PgC) estimates for each scenario.

This approach led to estimates of total BA, median FS, and median FI under modern conditions of a similar magnitude to the original GLM models and other global estimates (Andela et al., 2019; Humber et al., 2019) (Table 2).

Topographic and lightning variables were assumed not to change between the LGM and the present day. We used modern values, extrapolated out onto the exposed shelves, for the LGM experiments. The GLMs (Haas et al., 2022) include predictors associated with human activity, specifically human population density, road density and cropland cover. Population density is used as a measure of potential human ignitions and road density and

cropland cover as measures of landscape fragmentation. Including these anthropogenic predictors in the GLM
models was found to be essential to capture the global drivers of the observed spatial patterns of wildfires (Haas
et al., 2022). This is because modern fire regimes are influenced by human activity at a global scale (e.g. Marlon
et al., 2008; Bowman et al., 2020; Harrison et al., 2021). However, although the practice of foraging for plants by
some hunter-gatherer communities at the LGM has been shown (Liu et al., 2013), we presume that there was no
large-scale agriculture (or road networks) at the LGM. Additionally, information about pre-agricultural population
sizes is limited and highly uncertain (see e.g. Williams et al., 2013; Gautney & Holliday, 2015) and though some
regional models of human population do exist (Tallavaara et al., 2015), a reliable global product is not yet
available. To avoid confounding effects due to the high uncertainty of human impacts on global wildfire regimes,
we decided to exclude these anthropogenic predictors in all the experiments by setting them to zero. This ensured
that differences between the experiments were driven solely by climate and $CO_2$. We performed sensitivity
analysis to examine the impact of setting human predictors to zero under modern and LGM conditions (see S2).
Whilst BA and FS increase in the modern sensitivity analyses (especially in areas with high road density and
cropland density such as Europe and India) the effect was negligible for FI, highlighting the sensitivity of BA and
FS to human activity. Under LGM conditions, the effect of including human population was negligible for all
three fire properties. This reflects the slight and localised human impact on the natural landscape at the LGM
(Black et al, 2007; Fuller et al., 2014; Portenga et al., 2016).

When modelled GPP values were 0, BA, FS and FI was automatically set to 0. Modelled BA values smaller
than 0.001 were assumed to imply no burning, thus under these conditions FS and FI were also assumed to be 0
since both GLM models were trained on data of existing fires (see S3).
The resulting BA, FS and FI anomalies refer to the difference between the MOD climate/MOD $CO_2$
experiment and the three other experiments since each experiment is considered to represent the long-term average
spatial pattern for each fire property under the set experimental conditions. We used the sensitivity experiments
to quantify the separate effects of $CO_2$ and climate on BA, FS and FI independently. We then used the realistic
experiments to identify which predictors were driving the largest change between MOD and the three LGM
scenarios by excluding one predictor at a time from the GLM models, re-running the LGM experiments and
identifying which excluded variable caused the greatest change in the BA, FS and FI MOD-LGM anomalies in
each grid-cell. Comparing these results to the BA, FS and FI MOD-LGM anomalies of the full GLM models
allowed us to determine if the predictor was responsible for an increase or a decrease in BA, FS and FI.
We also compared the spatial patterns of BA, FS and FI with sedimentary charcoal data from the Reading
Palaeofire Database (RPD; Harrison et al., 2022). Sedimentary charcoal records provide a record of fire activity
but may reflect changes in both burnt area or completeness of combustion (Power et al., 2008) so this comparison
allowed us firstly to establish which of the fire regimes properties was most closely reflected in these records and
secondly which of the scenarios produced the most realistic patterns of burning. Model outputs and the charcoal
records were re-gridded to the coarsest resolution of the three climate models (2.5° x 1.875° resolution). We
calculated the number of correctly predicted BA, FS or FI anomalies (same sign within a given grid-cell),
separating positive and negative BA, FS or FI anomalies to assess the rate of false positives as well as false
negatives for each scenario and each LGM climate scenario.

## 3. Results

Global BA was substantially reduced compared to the realistic MOD scenario under all three realistic LGM scenarios, decreasing by 72% for the coldest CESM1.2 LGM scenario, 62% for the MPI-ESM1.2 LGM scenario and 41% for the warmest AWIESM1 LGM scenario. The largest decreases were observed in sub-Saharan Africa (excluding the tropical regions) as well as northern Australia and the Indian subcontinent (MPI-ESM1.2 and CESM1.2 LGM scenarios). Some increases in BA were observed in Alaska (MPI-ESM1.2 and AWIESM1 LGM scenarios) as well as south-East Asia, Indonesia, Papua-New-Guinea, and the northern tip of Australia. Increases in Somalia and Central America were also observed (MPI-ESM1.2 and AWIESM1 LGM scenarios). The number of grid cells (excluding ice covered cells) in which no burning occurred was 3 times higher in the MPI-ESM1.2 and AWIESM1 LGM scenarios and 4 times higher in the CESM1.2 LGM scenario compared to the realistic MOD scenario. This was driven by the expansion of desert and tundra biomes at the LGM. The Arabian plate, Middle East, inland China and Australia, and the tips of South America and Africa saw burning reduced to zero. Nearly all burning above 60°N was excluded, except for Alaska under the MPI-ESM1.2 and AWIESM1 LGM scenarios, with the exclusion extending down to 50°N for the CESM1.2 LGM scenario (see S3).

Globally, there was a large decrease in global median FS and FI when considering all grid-cells (not covered in ice) because of overall global reduction in burning. Under all three LGM scenarios, global median FS and FI were reduced to 0 compared to ~5km$^2$ for FS and 40W.km$^2$ for FI. However, when excluding grid-cells in which no burning occurred, both global median FS increased compared to the realistic MOD scenario (by ~16% under the two less conservative scenarios (MPI-ESM1.2 and AWIESM1) and by 12% under the CESM1.2 LGM scenario). The main increases in FS occurred in the Central America, Amazonia, tropical Africa as well as the Indian Subcontinent and Europe and Asia between 30°N and 60°N (except for CESM1.2 which had very few positive FS anomalies). The largest reductions were observed North America, southern Australia, Middle East, and the rest of Eurasia. Global median FI also increased in regions that were burning under two of the LGM scenarios, by 11% under the CESM1.2 LGM scenario and by 4% for MPI.ESM1.2 LGM scenario. Under the AWIESM1 LGM scenario global median FI decreased by 2% even when excluding grid-cells that were not burning. Despite this, changes in FI were spatially consistent across all three LGM scenarios, with increases in FI occurring primarily across the American and African continents, as well as the Mediterranean Basin and Europe and decreases occurring in Asia and inland Australia.

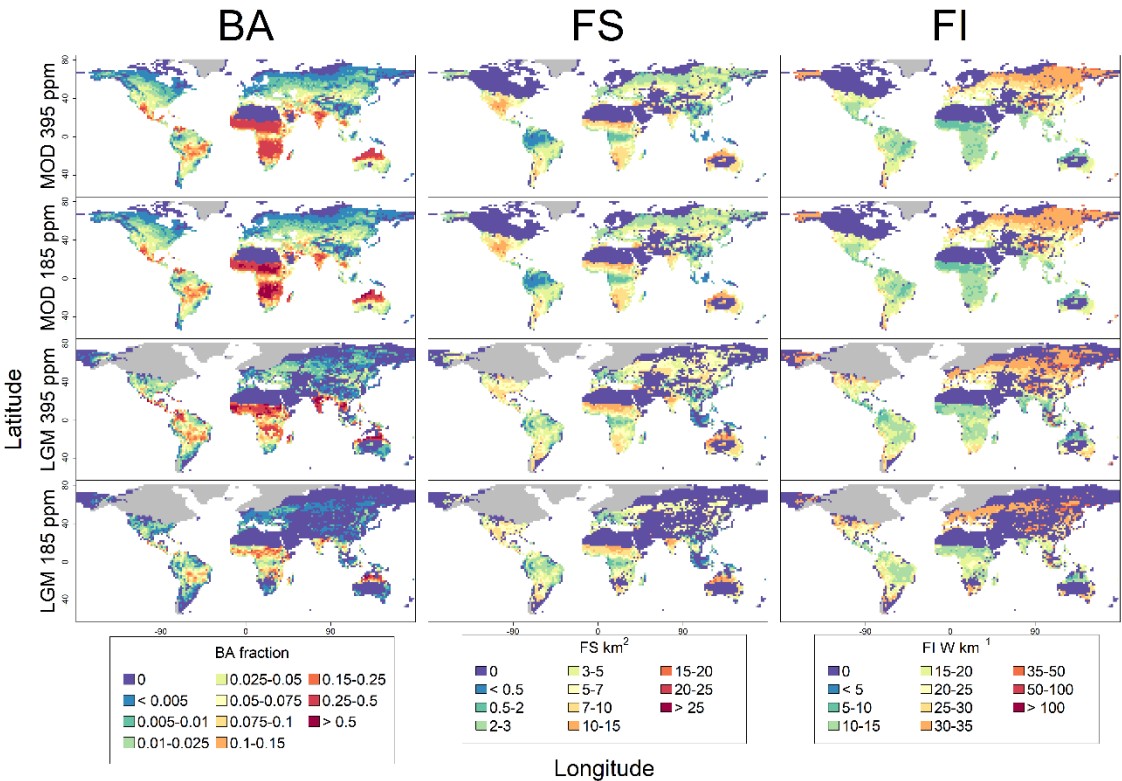

**Figure 3.** Experiments for BA, FS and FI for MPI-ESM1.2 LGM scenario (MOD 395 ppm and LGM 185 ppm represent the realistic modern-day simulation and LGM simulation, whilst MOD 185 ppm and LGM 395 ppm represent the $CO_2$ and climate sensitivity experiments respectively. The ice is shown in grey). (The other experiments can be found in S3)

Under low $CO_2$ levels with MOD climate (MOD climate/LGM $CO_2$) global BA decreased by ~ 70% under all three LGM scenarios (72% for CESM1.2 and AWIESM1, 73% for MPI-ESM1.2). Despite larger global decreased BA compared to the realistic LGM scenarios, the number of grid cells in which no burning occurred was only 1.7 times higher for MPI-ESM1.2 and AWIESM1 LGM scenarios and 1.5 times CESM1.2 LGM scenario compared to the realistic MOD scenario. The spatial pattern was consistent across all three LGM scenarios, with very few grid-points showing a positive BA anomaly relative to the MOD experiment. Though FS increased slightly under this sensitivity experiment when burning did occur, this increase was concentrated in the tropical regions of South America and Africa (mainly Amazonia), (except for AWIESM1 were increases were observed across Eurasia). In burning grid-cells, global median FI increased by ~ 15-18% in this sensitivity experiment (18% for MPI-ESM1.2 and CESM1.2, and 15% for AWIESM1). This spatial pattern was also consistent as with BA, with very few negative FI anomalies, except for regions ~ 20-30°N and ~20-30°S.

Under MOD $CO_2$ and LGM climate, BA decreased by 41% compared to the MOD experiment for the CESM1.2 LGM scenario and by 4% for the MPI-ESM1.2 LGM scenario but increased by 48% for the AWIESM1 LGM scenario, showing a strong sensitivity to climate. The number of grid cells in which no burning occurred was of similar amplitude to the previous sensitivity experiment for the MPI-ESM1.2 and AWIESM1 LGM scenarios (~1.8 times higher compared to the realistic MOD scenario) but was much higher for the CESM1.2 LGM scenario (~3.5 increase). When burning occurred, the global median FS increased under all LGM scenarios

by 17% for CESM1.2, 25% for MPI-ESM1.2 and 23% for AWIESM1. These increases were concentrated in tropical Africa, central America, and Russia, with decreases shown in North America and South Africa. Global median FI also increased under this sensitivity experiment by 2-3% for AWIESM1 and MPI-ESM1.2 but decreased by 5% for CESM1.2 LGM scenario, with decreases concentrated in Eurasia and North America.

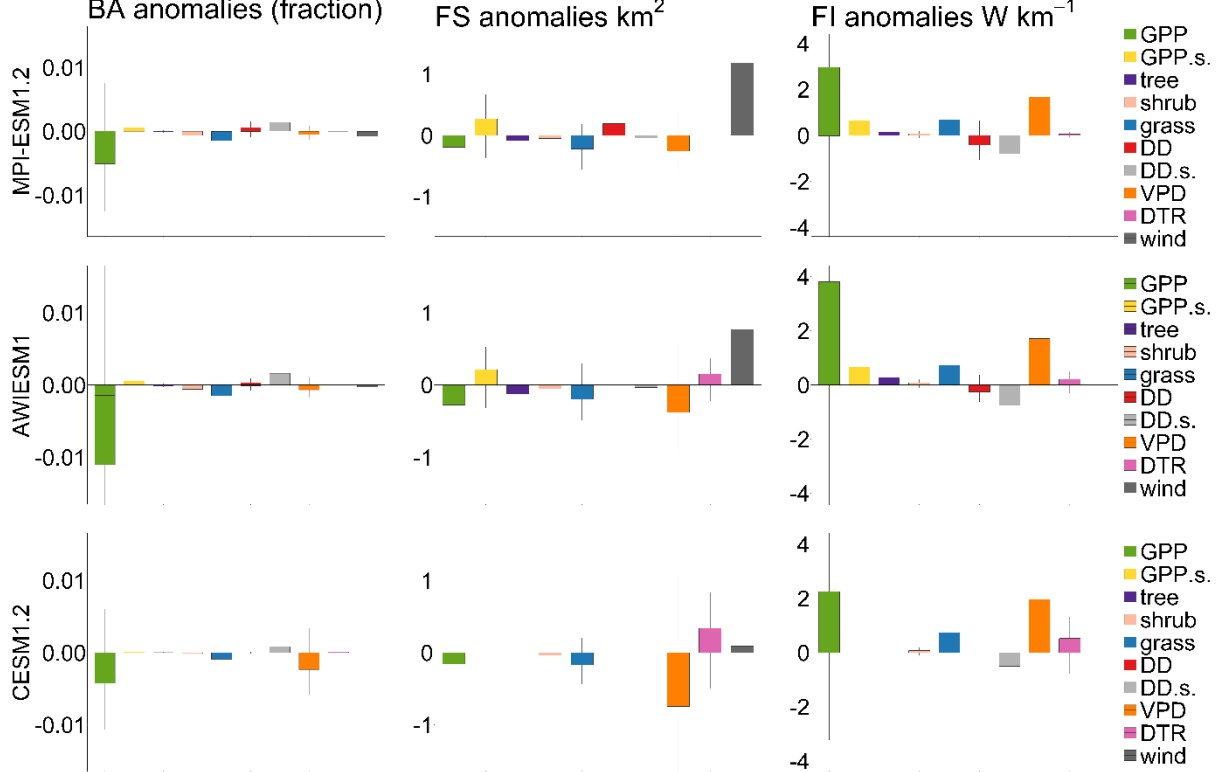

**Figure 4.** Boxplots showing relative importance of each predictor (GPP: gross primary production, GPP.s.: GPP seasonality, tree; tree cover, shrub; shrub cover, grass; grass cover, DD: dry days, DD.s.: dry days seasonality, VPD: vapour pressure deficit, DTR: diurnal temperature range, wind: wind speed) in driving the BA, FS or FI anomaly between the MOD 395 ppm and LGM 185 ppm experiment. For each grid cell common to both experiments (on modern-day continental shelves and masking the LGM ice sheets), the predictor which caused the largest change in the anomaly between the two experiments when it was excluded from the GLM model was retained, it is the change in anomaly that is shown here. This was taken as an indicator of relative importance of that predictor in driving the observed change for (a) the AWIESM1 LGM scenario, (b) the MPI-ESM-1.2 LGM scenario and (c) the CESM1.2 LGM scenario. A positive anomaly indicates the variable caused an increase in BA, FS or FI at the LGM and a negative anomaly indicates the variable caused a decrease in BA, FS or FI at the LGM.

Reductions in BA between the MOD and LGM scenarios were driven primarily by changes in GPP, grass cover, VPD and to a lesser extent dryness (dry days (DD) and dry-day seasonality (DD.s). Changes in FI were driven by changes in GPP as well as VPD, with changes in GPP seasonality also leading to increased FI in inland regions, reflecting both changes in climate and $CO_2$ levels for BA and FI. Increased FS was largely driven by increased wind speeds, as well as DD and diurnal temperature range (DTR) reflecting a strong climate effect as

well as GPP seasonality. Decreases in FS driven were by changes in GPP and grass cover, as well as VPD under
the CESM1.2 LGM scenario and DTR under the AWIESM1 LGM scenario (Figure 4). Changes in GPP and grass
cover were responsible for the largest reductions in burning, with these vegetation effects concentrated across
Africa and much of Eurasia (see Figure 5). In Amazonia, changes in DD were the most important factor, reducing
BA and FS (except for MPI-ESM1.2 which saw increased FS driven by DD). Increased BA in western Alaska
was driven by GPP in the MPI-ESM-1.2 and AWIESM1 LGM scenarios. Increased BA in tropical regions were
driven by grass cover, GPP and DD changes. Changes in VPD across the northern latitudes, especially of north
America and Europe, led to decreased BA in the most conservative CESM1.2 LGM scenario. FS decreased across
the Americas and Eurasia in the CESM1.2 LGM scenario because of low VPD values which reduced the
occurrence of burning and offset the increases caused by wind speed and DTR in the other two LGM scenarios.
Low values of VPD drove increases in FI across eastern North America, South America, western Africa, and
South-East Asia.

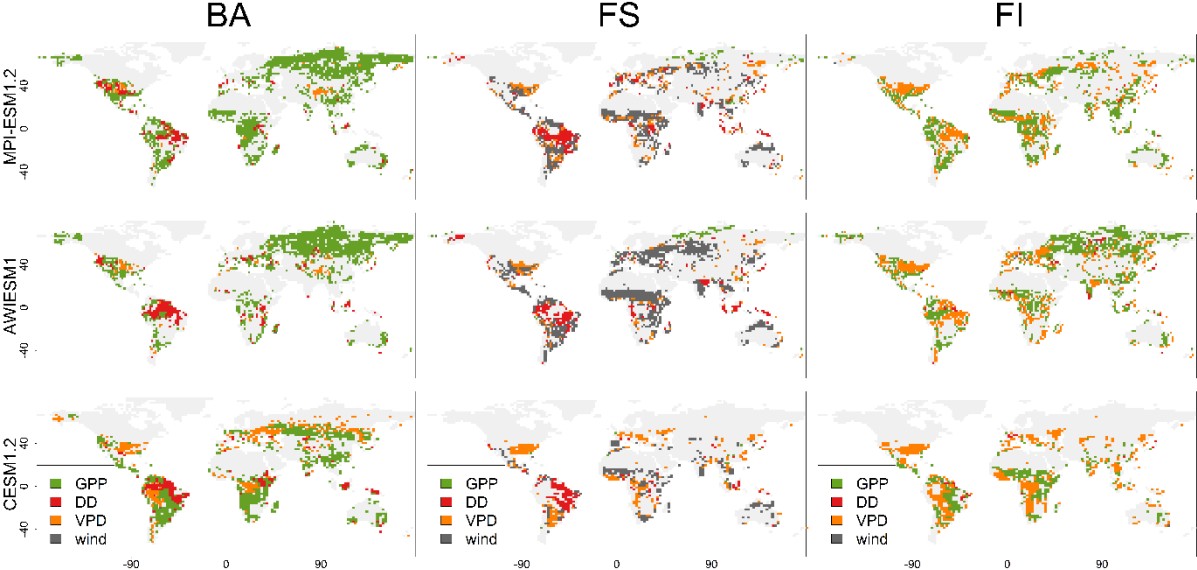


**Figure 5.** Map showing selection of four variables (GPP in green, DD in red, VPD in orange and wind in grey)

responsible for some of the most important grid-cell drivers in reducing BA, increasing FS and FI for (a)

AWIESM1 LGM scenario, (b) MPI-ESM1.2 LGM scenario and (c) CESM1.2 LGM scenario. Maps of most

important grid-cell drivers for all variables and all experiments can be found in S3.


Comparing the spatial patterns of the simulated BA anomalies with charcoal-based reconstructions of

the sign of changes in biomass burning (RPD; Harrison et al., 2022) showed that the best overall match occurred
when both the climate and $CO_2$ effect were considered, with a success rate of ~ 39-45% depending on the climate
scenario. The MPI-ESM1.2 and AWIESM1 LGM scenarios produced the best overall matches. None of the MOD
climate/LGM $CO_2$ experiments identified any of the positive BA anomalies shown by the charcoal records. The
LGM climate/MOD $CO_2$ experiments identified around half (~ 10-17%) of the negative BA anomalies identified
by the realistic experiment (17-20%) and the MOD climate/LGM $CO_2$ sensitivity experiment, and only performed
marginally better than the realistic experiment in identifying the positive BA anomalies (Table 3). Thus, although
this sensitivity experiment produced a similar overall agreement with the reconstructions as LGM climate/LGM
$CO_2$ simulations, only the realistic scenarios produced similar success rates for both the negative and positive BA
anomalies. Climate change alone produced too few negative anomalies matches; $CO_2$ changes alone resulted in
no positive anomaly matches.

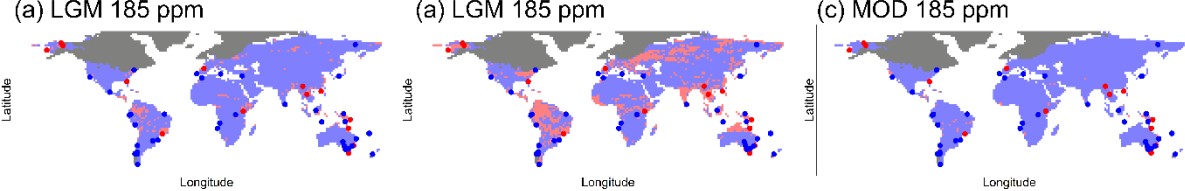

**Figure 6.** Comparison of BA anomalies between the experiment outputs from the MPI-ESM1.2 LGM scenario
with charcoal records from the Reading Palaeofire Database (RPD) for (a) the realistic LGM experiment (b) the
LGM climate/MOD $CO_2$ sensitivity experiment and (c) the MOD climate/LGM $CO_2$ sensitivity experiment The
modelled positive LGM-MOD anomalies are shown in red and LGM-MOD negative anomalies in blue. Dotted
red (positive anomaly) and blue (negative anomaly) points show the location of the RPD records for the LGM.
The LGM ice sheets are shown in dark blue.

| BA experiments | | MPI_ESM1.2 | | | AWIESM1 | | | CESM1.2 LGM | | |
|---|---|---|---|---|---|---|---|---|---|---|
| Scenario | | LGM | MOD | LGM | LGM | MOD | LGM | LGM | MOD | LGM |
| | RPD | 190 | 190 | 395 | 190 | 190 | 395 | 190 | 190 | 395 |
| Negative RPD anomalies | | | | | | | | | | |
| Number of records | 35 | 20 | 21 | 13 | 17 | 21 | 10 | 20 | 20 | 17 |
| Successful identification (percentage) | | 57 | 60 | 37 | 49 | 60 | 29 | 57 | 57 | 49 |
| Positive RPD anomalies | | | | | | | | | | |
| Number of records | 16 | 3 | 0 | 8 | 6 | 0 | 5 | 0 | 0 | 3 |
| Successful identification (percentage) | | 19 | 0 | 50 | 38 | 0 | 31 | 0 | 0 | 19 |
| Total RPD anomalies | | | | | | | | | | |
| Number of records | 51 | 23 | 21 | 21 | 23 | 21 | 15 | 20 | 20 | 20 |
| Successful identification (percentage) | | 45 | 41 | 41 | 45 | 41 | 29 | 39 | 39 | 39 |

**Table 2.** Comparison of sign in BA anomalies (between the MOD climate/MOD $CO_2$ experiment and other
three experiments respectively) at the location of each RDP charcoal-based reconstruction record. A positive
anomaly represents increased biomass burning, and a negative anomaly represents decreased biomass burning.
A successful identification means that the sign of the experiment anomaly and the sign of the RPD charcoal-
based reconstructions are the same.
The sign of the charcoal records could reflect changes in FS or FI as well as BA. However, the success rates
in predicting the sign of the charcoal anomalies (both positive and negative) were not as good for FS (27-31%)
and FI (24-30%) than those obtained for BA for the realistic LGM experiment. Furthermore, both FS and FI did
not perform any better than BA under any experiment, with the sensitivity experiments matching the charcoal
anomalies slightly better for FS and FI than the realistic LGM experiment (see S4).
**4.    Discussion**

Our simulations show a global reduction in burning at the LGM but increased median fire size and

intensity when burning did occur. BA, FS and FI were all sensitive to changes in vegetation driven directly by
$CO_2$ levels alone. BA and FI were most sensitive to this effect, with the climate effect dampening the effect of
$CO_2$ alone when both are included. The largest reductions in burning occurred when only the $CO_2$ effect was
considered although this experiment had fewer regions in which burning was excluded completely. This suggests
that the reduction in burning was more spatially consistent and widespread under these conditions than when both
effects were accounted for. The sensitivity of BA to $CO_2$ is explained by the reduction in fuel availability under
low $CO_2$, a strong constraint on burnt area. For FI, including a $CO_2$ effect also amplified the overall global signal.
This $CO_2$ effect is most likely driven by the negative relationship between GPP and FI fitted by the empirical
model. Whilst this relationship might seem counter-intuitive, it has a sound basis. The most intense fires occur in
regions with a seasonal variation in productivity rather than the most productive environments such as tropical
forests (Archibald et al., 2013). High productivity can (under some climate conditions) increase the frequency of
burning, which also reduces fuel loads (Rodrigues et al., 2019). Under appropriate climate conditions, there can
be long-term fuel build-up in areas of low productivity that is not offset by frequent burning. All these factors
help to explain why FI is not reduced at the LGM when burning occurs even though BA is. Low $CO_2$ decreased
FS except for tropical regions and reduced the impact of climate in the realistic scenarios. We hypothesize this is
because of decreased productivity leading to patchier vegetation, and hence reduced fuel continuity, which is a
factor limiting wildfire spread (Dial et al., 2022; Schertzer et al., 2015).

Changes in climate alone also affected all three modelled wildfire properties. The climate effect was

larger than the $CO_2$ effect across all models for FS, with increases in wind, DD and DTR driving the change. BA
was particularly sensitive to the amplitude of climate change: climate change alone greatly reduced BA under the
coldest LGM scenario (CESM1.2), had a limited effect in the intermediate LGM scenario (MPI-ESM1.2) and
increased BA in the warmest LGM scenario (AWIESM1). The amplitude of change in VPD, a measure of
atmospheric moisture, relative to other climate variables was especially important in influencing overall trends.
In the case of BA, large decreases in VPD under the CESM1.2 climate scenario led to much more substantial
reductions, most likely due to an increase in fuel moisture. Additionally, though stronger winds and increased
DTR were the main drivers of larger wildfires at the LGM, low VPD values in CESM1.2 severely limited FS and
FI in the northern latitudes. VPD has been shown to influence wildfire ignition and wildfire spread (Sedano &
Randerson, 2014), and our results suggest high atmospheric moisture can inhibit fire spread. When vegetation
was sufficiently abundant however, low VPD values were key in driving intensity. Although vegetation
productivity was lower at the LGM, decreased VPD may have contributed to larger fuel build-ups, thus increasing
fuel loads. This highlights the sensitivity of the fire regime not just to overall climate change but the relative
amplitude of change in individual climate variables.

Our model results reproduce the global reduction of biomass burning at the LGM observed from ice

cores and sedimentary charcoal records (Daniau et al., 2012; Harrison et al., 2022; Power et al., 2008; Rubino et
al., 2016). Some studies have indicated the occurrence of high-intensity wildfires on the Palaeo-Agulhas Plain of
South Africa, tropical regions, northern Australia, and central China at the LGM (Kraaij et al., 2020; Power et al.,
2008; Rowe et al., 2021; Ruan et al., 2020; M. Song et al., 2023). Our results are consistent with the trends in
these regions. The LGM simulations of BA that account for both climate and $CO_2$ appear to fit the charcoal records
best. The spatial patterns of BA at the LGM were more consistent with the patterns shown by sedimentary charcoal
records than FS and FI, consistent with the assumption that charcoal abundance can be used as a measure of
biomass burning. The FS and FI anomaly patterns were less consistent than that of BA, suggesting a regime of
less burning but larger and more intense wildfires at the LGM could be consistent with the charcoal records.
Whilst FI has been reconstructed from charcoal (e.g. Duffin, 2008; Snitker, 2018) there are currently no
comparable measures that record FS or FI changes globally. Charcoal records are not available from some regions,
further limiting our ability to evaluate the models, particularly in Eurasia and inland South America where low
$CO_2$ leads to large reductions in BA that are not observed when only climate is considered.
Our results are based on simple empirical models for BA, FS and FI. However, the inferred changes in
BA are like those of Martin Calvo et al. (2015) who used the Land surface Processes and eXchanges (LPX)
dynamic global vegetation model. Empirical models have been shown to perform as well as more complex
process-based models in simulating burned area under modern-day conditions (Hantson et al., 2020). Thus, our
conclusions about the relative impact of climate and $CO_2$ changes on fire properties are unlikely to be adversely
affected by the relative simplicity of the models used. Their simplicity facilitates running multiple scenarios and
diagnosis of the factors influencing changes in wildfire properties.
The effect of human activity was not considered in this analysis and as such no conclusions can be drawn
on how human activity may affect these trends. Although this is a limitation, we believe it is unlikely that human
activity would substantially impact the response of wildfire regimes to the changes in climate and $CO_2$ observed
here. Pre-agricultural hunter-gatherer populations used fire for land management, for example to facilitate hunting
and to promote the local abundance of food plants (Bowman, 1998; Gott, 2005), although recent work indicates
that the burning regimes they practiced tended to reduce fire overall compared to the natural state (see e.g.
Constantine IV et al., 2023). However, the areas suitable for hunter-gatherer populations was much reduced at the
LGM by generally colder and drier climates and hunter-gatherer populations were confined to climatically suitable
refugia (see e.g. Williams et al., 2013; Blinkhorn et al., 2022). Furthermore, although the estimates of population
density are highly uncertain, the LGM population of Australia was less than 5% of the modern population and the
reduction in Africa was even larger (Gautney and Holliday, 2015). Palaeoecological evidence from Australia
suggests that the use of fire by pre-agricultural hunter-gatherers had a low impact on the environment before the
late Holocene (e.g. Black et al., 2007; Fuller et al., 2014; Portenga et al., 2016). Thus, it is unlikely that human
activities during the LGM would have substantially increased fire or offset the impact of the changes in climate
and $CO_2$ on fire regimes. Previous studies show a weak influence of population and land-use change on driving
global wildfire trends prior to the 18th century (e.g Pechony and Shindell, 2010; Bowman et al., 2020) and a sharp
human-driven decline in wildfire activity since the mid-ninetieth century (e.g Marlon et al., 2008; Wang et al.,
2010). This recent reduction in global biomass burning was most likely driven by population growth and land-use
change leading to increased landscape fragmentation, which tends to suppress fire spread (e.g. Knorr et al., 2014;
Andela et al., 2016; Harrison et al., 2021).
These results add to a growing body of literature highlighting the importance of considering not only
changes in wildfire weather but also vegetation properties in projections of future wildfire regimes (e.g. Harrison
et al., 2021; Kuhn-Régnier et al., 2021; Pausas & Keeley, 2021). The impact of rising $CO_2$ levels will most likely
enhance vegetation growth and litter accumulation, which are important controls on fuel availability, continuity,
and load. However, climate and specifically VPD may have opposing effects to that of rising $CO_2$ levels. Since

VPD controls plant growth, increasing VPD can limit ecosystem productivity and tree growth, in turn reducing fuel loads (Williams et al. 2013). Nevertheless, VPD has also been shown to increase litter fall, thus increasing available dead fuel (Resco de Dios 2020, De Faria et al. 2017). As such, it is important to consider how temporal and spatial scales affect the response of vegetation to changing VPD (Grossiord et al., 2020). Although the trade-offs between future increases in $CO_2$ and reductions in productivity due to higher temperatures and atmospheric dryness are not fully understood, this work highlights the importance of considering both. These effects will most likely not be evenly distributed across the globe (Gonsamo et al., 2021; Piao et al., 2020; van der Sleen et al., 2015) and $CO_2$ effects may be more important in some regions than others. In fuel-limited ecosystems, $CO_2$ fertilization could increase fuel loads and fuel continuity, increasing overall burnt area but also the potential for larger and more intense wildfires. This is particularly worrying in regions with anticipated decreases in atmospheric moisture, especially since evidence suggests rising VPD may only counteract a small proportion of $CO_2$-induced plant growth (Y. Song et al., 2022). Increased woody thickening, for example in tropical South Asia (Kumar et al., 2021; Scheiter et al., 2020), may also alter fuel loads in regions that are likely to be vulnerable to ignition under a drier and warmer atmosphere (Clarke et al., 2022). Whilst climate variables such as DD and DTR have also shown to be strong controls of global wildfires regimes (e.g. Bistinas et al., 2014; Forkel et al., 2019; Kuhn-Régnier et al., 2021), this study highlights the importance of VPD relative to other climate variables in driving spatial patterns of BA, FS and FI. This is in line with previous studies that have highlighted the important role of VPD in promoting fuel loads and fire spread (e.g. Diffenbaugh et al., 2021; Grillakis et al., 2022; Duane et al., 2021; Balch et al., 2022). Correctly projecting changes in fuels in the next century will require considering both the effect of VPD and effects of $CO_2$ on plant growth and fuel loads.

Our results stress the importance of accounting for the effects of $CO_2$ on vegetation when considering how future fire regimes may evolve. Different aspects of the fire regime respond differently to changes in fuel properties. Without accounting for this crucial effect, our understanding of future risks will remain limited.

**Code availability.** All code used in this paper is available at freely available for use in RStudio: the code for the GLM models is available at https://doi.org/10.6084/m9.figshare.19071044.v1, and the code to generate the experiments are available at: https://doi.org/10.6084/m9.figshare.22285303.v2 and https://doi.org/10.6084/m9.figshare.22285279.v2.

**Data availability:** All LGM data can be retrieved from https://esgf-node.llnl.gov/projects/cmip6/, all modern data can be retrieved from references provided. The P Model documentation is available at https://pyrealm.readthedocs.io/en/latest/ and the BIOME4 documentation is available at https://pmip2.lsce.ipsl.fr/synth/biome4.shtml and https://github.com/jedokaplan/BIOME4.

**Author contributions.** Experiments conception, strategy and interpretation were developed by O H, ICP and SPH jointly. OH performed the data processing and analysis, and produced the graphics and Tables. OH wrote the original draft; SPH and ICP contributed to the final draft.

**Competing interests**. The contact author has declared that neither themselves nor any other authors have a conflict of interest.

**Acknowledgements and financial support.** OH acknowledges support from the NERC Centre for Doctoral Training in Quantitative and Modelling skills in Ecology and Evolution (Grant No. NE/S007415/1) and from the Leverhulme Trust through the Leverhulme Centre for Wildfires, Environment and Society (Grant No. RC-2018-023). Special thanks to David Orme for this help with setting up BIOME4. ICP acknowledges support from the European Research Council (787203 REALM) under the European Union's Horizon 2020 research programme. SPH is supported by the European Research Council (694481 GC2.0) under the same programme. This work is a contribution to the LEMONTREE (Land Ecosystem Models based On New Theory, obseRvations and ExperimEnts) project, funded through the generosity of Eric and Wendy Schmidt by recommendation of the Schmidt Futures program.

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
