# Peer review of "The response of wildfire regimes to Last Glacial Maximum carbon dioxide"

_EGUsphere, 2023_

## Author Response (AR1)

**Editor's comment:** "Based on the reviewers' comments and your replies, I have decided that the manuscript requires major revision.

Besides other comments, I want to highlight the comment on the lack of human activity data raised by referee #2. I fully support this claim, although it might be problematic to separate inhabited regions from uninhabited and run the analysis separately. I am still convinced that human impact could significantly affect the fire even during the LGM in many regions (e.g. Europe, with an estimated habitable area of 36 %, Tallavaara et al. 2015). You also expect no agriculture during the LGM, but this is quite a general statement. We have much earlier evidence than 12000 BP for intentional plant use and land management, e.g. in China (Liu et al. 2013). To substantiate your claim that human impact can be completely neglected, you could still design a test with a limited number of localities."

To assess the potential human impact on fire regimes at the LGM we have run three additional sensitivity analysis including human population for Europe, Africa, and Australia under realistic LGM conditions.

We ran regional LGM analysis with human population (LGM climate/LGM $CO_2$ popd) and without (LGM climate/LGM $CO_2$) and compared the amplitude of change between these two experiments with the amplitude of change between the realistic LGM experiment (LGM climate/LGM $CO_2$) and the realistic modern experiment (MOD climate/MOD $CO_2$). This allowed us to assess whether setting human population densities to zero had a significant impact on the LGM and whether it had the potential to influence the global trends between the LGM and the MOD experiments. For the European experiment, we used the gridded data produced by Tallavaara et al., (2015). For the African and Australian experiments, we used the estimated population densities from Gautney and Holliday (2015) for areas that were considered habitable and set the population density to zero in areas considered uninhabitable. We considered an area uninhabitable when it was a modelled as desert or barren by BIOME4 (Kaplan et al., 2003) or was at an altitude above 3000 m, following the methodology by Gautney & Holliday (2015). We compared our total and habitable areas to the estimates of Gautney and Holliday (2015). Although there were some differences, we believe that our estimates are fairly similar (though slightly higher in Africa and slightly lower in Australia).

| | Africa | | | | Australia | | | |
|---|---|---|---|---|---|---|---|---|
| | Total land area (km²) | Habitable (km²) | % Habitable | Number of people | Total land area (km²) | Habitable (km²) | % Habitable | Number of people |
| AWIESM1 | 34,028,261 | 20,982,697 | 62% | 2,566,184 | 9,456,315 | 7,650,814 | 81% | 38,254 |
| MPI-ESM1.2 | 34,071,392 | 21,553,863 | 63% | 2,636,037 | 9,456,315 | 7,650,814 | 81% | 38,254 |
| CESM1.2 | 34,322,097 | 22,199,317 | 65% | 2,714,976 | 9,421,515 | 7,763,107 | 82% | 38,816 |
| Gautney & Holliday (2015) | 30,493,900 | 12,846,597 | 42% | 1,571,139 | 11,021,024 | 9,418,730 | 85% | 47,093 |

**Table 1.** Habitable land area and population estimates for Africa and Australia

[Figure]

**Figure 1.** Maps of deserted areas at the LGM (shown in red) (a) showing the extent of the Sahara and Arabian Deserts according to Gautney & Holliday (2015), (b) showing the Great Victorian Desert, the Simpson Desert, and the Great and Little Sandy Deserts according to Gautney & Holliday (2015), (c) showing the extent of desert and barren simulated by BIOME4 for Africa and (d) showing the extent of desert and barren areas simulated by BIOME4 for Australia

Although some hunter-gatherer communities foraged for plants at the LGM (Liu et al., 2013), there is large uncertainty surrounding the extent of this practice at a global scale. Additionally, cropland in the GLMs is understood as a large-scale landscape feature, significant at least a ~ 55km resolution at the equator. It is reasonable to assume that hunter-gatherer communities at the LGM did not cultivate crops on this scale. Roads and crop cover were therefore set to zero under all LGM experiments, including the experiment with human population estimates.

We observed very slight differences between the regional LGM experiments when human population densities were included and when they were not (less than 5% change for BA, FS and FI). These differences were much smaller than the differences between the LGM experiment and the MOD experiment (see Table 2).

| | AWI-ESM1 | | | MPI-ESM1.2 | | | CESM1.2 | | |
|---|---|---|---|---|---|---|---|---|---|
| | Europe | Africa | Australia | Europe | Africa | Australia | Europe | Africa | Australia |
| **Burnt area (km$^2$)** | | | | | | | | | |
| MOD climate/MOD CO$_2$, | 101,041 | 6,568,740 | 1,491,914 | 99,222 | 6,439,953 | 1,441,669 | 97,967 | 6,323,321 | 1,490,743 |
| LGM climate/LGM CO$_2$ | 20,124 | 3,099,159 | 2,108,874 | 22,494 | 1,410,410 | 1,135,466 | 10,409 | 1,534,357 | 886,142 |
| LGM climate/LGM CO$_2$ popd | 20,210 | 3,114,700 | 2,113,961 | 22,603 | 1,417,576 | 1,138,501 | 10,500 | 1,542,090 | 890,406 |
| % change between MOD and LGM | − 80 | − 53 | 41 | − 77 | − 78 | − 21 | − 89 | − 76 | − 41 |

| | | | | | | | | | |
|---|---|---|---|---|---|---|---|---|---|
| % change between LGM and LGM popd | 0.43 | 0.50 | 0.24 | 0.48 | 0.27 | − 0.85 | − 0.85 | 0.50 | 0.48 |
| **Fire size (km$^2$)** | | | | | | | | | |
| MOD climate/MOD CO$_2$, | 5.12 | 8.88 | 11.07 | 4.97 | 9.33 | 11.96 | 6.07 | 9.22 | 12.79 |
| LGM climate/LGM CO$_2$ | 6.51 | 9.05 | 12.93 | 5.97 | 7.98 | 11.87 | 7.32 | 7.52 | 13.72 |
| LGM climate/LGM CO$_2$ popd | 5.51 | 9.05 | 12.93 | 5.97 | 7.98 | 11.87 | 7.51 | 7.52 | 13.72 |
| % change between MOD and LGM | 27 | 2 | 17 | 20 | 14 | 0.75 | 21 | − 18 | 7 |
| % change between LGM and LGM popd | 0 | 0 | 0 | 0 | 0 | 0 | 0 | 0 | 0 |
| **Fire intensity (W.km$^{-1}$)** | | | | | | | | | |
| MOD climate/MOD CO$_2$, | 29.97 | 16.65 | 20.80 | 30.43 | 16.48 | 20.09 | 27.19 | 16.74 | 19.63 |
| LGM climate/LGM CO$_2$ | 37.66 | 17.90 | 18.14 | 37.16 | 20.56 | 21.81 | 41.39 | 23.68 | 29.89 |
| LGM climate/LGM CO$_2$ popd | 37.62 | 17.81 | 18.05 | 37.13 | 20.46 | 21.71 | 44.61 | 23.53 | 29.75 |
| % change between MOD and LGM | 26 | 8 | − 9 | 22 | 25 | 9 | 52 | 41 | 52 |
| % change between LGM and LGM popd | − 0.1 | − 0.5 | − 4.5 | − 0.1 | − 0.5 | − 0.5 | − 0.5 | − 0.6 | − 0.5 |

**Table 1.** Regional BA, FS and FI estimates for MOD climate/MOD CO$_2$, LGM climate/LGM CO$_2$ and LGM climate/LGM CO$_2$ popd

[Figure]

**Figure 2.** Percentage change of BA, FS and FI when including population estimates at the LGM for Europe

[Figure]

**Figure 3.** Percentage change of BA, FS and FI between the realistic LGM experiment (LGM climate/LGM $CO_2$) and the the realistic modern experiment (MOD climate/MOD $CO_2$) for Europe

[Figure]

**Figure 4.** Percentge change of BA, FS and FI when including population estimates at the LGM for Africa

[Figure]

**Figure 5.** Percentage change of BA, FS and FI between the realistic LGM experiment (LGM climate/LGM CO$_2$) and the realistic modern experiment (MOD climate/MOD CO$_2$) for Africa

[Figure]

**Figure 6.** PercentAGE change of BA, FS and FI when including population estimates at the LGM for Australia

[Figure]

**Figure 7.** Percentage change of BA, FS and FI between the realistic LGM experiment (LGM climate/LGM CO$_2$) and the realistic modern experiment (MOD climate/MOD CO$_2$) for Australia

The modern human sensitivity runs (from original analysis) showed that setting human predictors to zero under modern conditions had no effect on FI but a strong effect on BA and FS, leading to large increases when human activity was "off" (see Table 3). However, this increase in BA and FS at the modern was driven by road density and to a lesser extent cropland, not by human population. The original GLM models are not very sensitive to human population alone (population density did not meet the minimum significance threshold for inclusion in the final models (Haas et al., 2021)), and we are

using realistic (and very low) population densities for the LGM, the small effects shown are not surprising. The limited impact of population density alone in driving global fire regimes in the GLM models is in line with research showing the importance of anthropogenic landscape fragmentation when modelling how humans influence fire regimes, as opposed to focusing solely on the effect of human population (e.g. Bistinas et al., 2014, Knorr et al 2014, 2016; Kelley et al., 2019; Harrison et al., 2021).

**Table 3.** Sensitivity of GLM models to human activity using both observations and BIOME4 derived vegetation and GPP

| Inputs for land cover and P Model GPP (Cucchi et al., 2020) | ESA CCI Landcover NASA/GIMS fAPAR 3g | BIOME4 (Kaplan et al., 2003) | Global estimates from the literature |
|---|---|---|---|
| **Burnt area (millions km²)** | | | |
| *Human activity on* | 4.42 | 4.25 | [1.87 – 4.6] (Humber et al., 2019) |
| *Human activity off* | 7.41 | 11.27 | |
| *% change* | 40.35 | 62.29 | |
| **Fire size (km²)** | | | |
| *Human activity on* | 3.36 | 3.61 | 4.4 (Andela et al., 2019) (does not include wildfires smaller than 0.21 km²) |
| *Human activity off* | 5.34 | 6.25 | |
| *% change* | 37.08 | 42.24 | |
| **Fire intensity (W.km⁻¹)** | | | |
| *Human activity on* | 40.00 | 31.41 | |
| *Human activity off* | 39.20 | 31.17 | |
| *% change* | − 2.04 | − 0.77 | |

The results of these sensitivity analysis, combined with the large uncertainty associated with human population numbers at the LGM, justify our approach of setting human predictors to zero. In doing so, we are not stating that the effect of human impact was negligible; rather that due to large uncertainties around human activity at the LGM, the most transparent approach is to run all the experiments in a counterfactual "human-less" world, both in the modern and the LGM in order to focus on the effects of climate and $CO_2$. The aim of this approach was to eliminate any confounding effect associated with human activity, such as the ones raised by referee #2. This allowed us to concentrate on the effects of LGM climate and $CO_2$ on global fire regimes.

We suggest creating a new supplementary section in which we include these human sensitivity analyses at the LGM as well as the human sensitivity analyses under modern conditions (which are currently included in the original manuscript) and demonstrate why our choice of setting human predictors to zero was made. Additionally, we suggest adding a section to the Discussion to highlight the limitation of excluding human effects from the analysis.

We suggest the following changes to the manuscript:

In the methods section, line 153: "The GLMs (Haas et al., 2022) include predictors associated with human activity, specifically human population density, road density and cropland cover. Population

density is used as a measure of potential human ignitions and road density and cropland cover as measures of landscape fragmentation. Including these anthropogenic predictors in the GLM models was found to be essential to capture the global drivers of the observed spatial patterns of wildfires (Haas et al., 2022). This is because modern fire regimes are influenced by human activity at a global scale (e.g. Marlon et al., 2008; Bowman et al., 2020; Harrison et al., 2021). However, although the practice of foraging for plants by some hunter-gatherer communities at the LGM has been shown (Liu et al., 2013), we presume that there was no large-scale agriculture (or road networks) at the LGM. Additionally, information about pre-agricultural population sizes is limited and highly uncertain (see e.g. Williams et al., 2013; Gautney & Holliday, 2015) and though some regional models of human population do exist (Tallavaara et al., 2015), a reliable global product is not yet available. To avoid confounding effects due to the high uncertainty of human impacts on global wildfire regimes, we decided to exclude these anthropogenic predictors in all the experiments by setting them to zero. This ensured that differences between the experiments were driven solely by climate and $CO_2$. We performed sensitivity analysis to examine the impact of setting human predictors to zero under modern and LGM conditions (see S2). Whilst BA and FS increase in the modern sensitivity analyses (especially in areas with high road density and cropland density such as Europe and India) the effect was negligible for FI, highlighting the sensitivity of BA and FS to human activity. Under LGM conditions, the effect of including human population was negligible for all three fire properties. This reflects the slight and localised human impact on the natural landscape at the LGM (Black et al, 2007; Fuller et al., 2014; Portenga et al., 2016)."

We also suggest moving table 2 to S2.

And in the discussion, line 377: "The effect of human activity was not considered in this analysis and as such no conclusions can be drawn on how human activity may affect these trends. Although this is a limitation, we believe it is unlikely that human activity would substantially impact the response of wildfire regimes to the changes in climate and $CO_2$ observed here. Pre-agricultural hunter-gatherer populations used fire for land management, for example to facilitate hunting and to promote the local abundance of food plants (Bowman, 1998; Gott, 2005), although recent work indicates that the burning regimes they practiced tended to reduce fire overall compared to the natural state (see e.g. Constantine IV et al., 2023). However, the areas suitable for hunter-gatherer populations was much reduced at the LGM by generally colder and drier climates and hunter-gatherer populations were confined to climatically suitable refugia (see e.g. Williams et al., 2013; Blinkhorn et al., 2022). Furthermore, although the estimates of population density are highly uncertain, the LGM population of Australia was less than 5% of the modern population and the reduction in Africa was even larger (Gautney and Holliday, 2015). Palaeoecological evidence from Australia suggests that the use of fire by pre-agricultural hunter-gatherers had a low impact on the environment before the late Holocene (e.g. Black et al., 2007; Fuller et al., 2014; Portenga et al., 2016). Thus, it is unlikely that human activities during the LGM would have substantially increased fire or offset the impact of the changes in climate and CO2 on fire regimes. Previous studies show a weak influence of population and land-use change on driving global wildfire trends prior to the 18th century (e.g Pechony and Shindell, 2010; Bowman et al., 2020) and a sharp human-driven decline in wildfire activity since the mid-ninetieth century (e.g Marlon et al., 2008; Wang et al., 2010). This recent reduction in global biomass burning was most likely driven by population growth and land-use change leading to increased landscape fragmentation, which tends to suppress fire spread (e.g. Knorr et al., 2014; Andela et al., 2016; Harrison et al., 2021)."

References:

Andela, N., Morton, D.C., Giglio, L., Chen, Y., van der Werf, G.R., Kasibhatla, P.S., DeFries, R.S., Collatz, G.J., Hantson, S., Kloster, S. and Bachelet, D., 2017. A human-driven decline in global burned area. *Science*, *356*(6345), pp.1356-1362.

Bistinas, I., Harrison, S.P., Prentice, I.C. and Pereira, J.M.C., 2014. Causal relationships versus emergent patterns in the global controls of fire frequency. *Biogeosciences*, *11*(18), pp.5087-5101.

Black, M. P., Mooney, S. D., & Haberle, S. G. (2007). The fire, human and climate nexus in the Sydney Basin, eastern Australia. *The Holocene*, 17(4), 469-480.

Blinkhorn, J., Timbrell, L., Grove, M., & Scerri, E. M. L. (2022). Evaluating refugia in recent human evolution in Africa. *Philosophical Transactions of the Royal Society B*, *377*(1849), 20200485.

Bowman, D. M. J. S. (1998). The impact of Aboriginal landscape burning on the Australian biota. *The New Phytologist*, *140*(3), 385–410.

Bowman, D. M. J. S., Kolden, C. A., Abatzoglou, J. T., Johnston, F. H., van der Werf, G. R., & Flannigan, M. (2020). Vegetation fires in the Anthropocene. *Nature Reviews Earth & Environment*, *1*(10), 500–515.

Constantine IV, M., Williams, A. N., Francke, A., Cadd, H., Forbes, M., Cohen, T. J., Zhu, X., & Mooney, S. D. (2023). Exploration of the burning question: a long history of fire in eastern Australia with and without people. *Fire*, *6*(4), 152.

Fuller, D. Q., Denham, T., Arroyo-Kalin, M., Lucas, L., Stevens, C. J., Qin, L., Allaby, R. G., & Purugganan, M. D. (2014). Convergent evolution and parallelism in plant domestication revealed by an expanding archaeological record. *Proceedings of the National Academy of Sciences*, *111*(17), 6147–6152.

Gautney, J. R., & Holliday, T. W. (2015). New estimations of habitable land area and human population size at the Last Glacial Maximum. *Journal of Archaeological Science*, *58*, 103–112.

Gott, B. (2005). Aboriginal fire management in south-eastern Australia: aims and frequency. *Journal of Biogeography*, 1203–1208.

Harrison, S. P., Prentice, I. C., Bloomfield, K. J., Dong, N., Forkel, M., Forrest, M., ... & Simpson, K. J. (2021). Understanding and modelling wildfire regimes: an ecological perspective. *Environmental Research Letters*, 16(12), 125008.

Liu, L., Bestel, S., Shi, J., Song, Y., & Chen, X. (2013). Paleolithic human exploitation of plant foods during the last glacial maximum in North China. Proceedings of the National Academy of Sciences of the United States of America, 110(14), 5380–5385. https://doi.org/10.1073/pnas.1217864110

Knorr, W., Jiang, L. and Arneth, A., 2016. Climate, $CO_2$ and human population impacts on global wildfire emissions. *Biogeosciences*, *13*(1), pp.267-282.

Knorr, W., Kaminski, T., Arneth, A. and Weber, U., 2014. Impact of human population density on fire frequency at the global scale. *Biogeosciences*, *11*(4), pp.1085-1102.

Marlon, J.R., Bartlein, P.J., Carcaillet, C., Gavin, D.G., Harrison, S.P., Higuera, P.E., Joos, F., Power, M.J. and Prentice, I.C., 2008. Climate and human influences on global biomass burning over the past two millennia. *Nature Geoscience*, 1(10), pp.697-702.

Tallavaara, M., Luoto, M., Korhonen, N., Järvinen, H. and Seppä, H., 2015. Human population dynamics in Europe over the Last Glacial Maximum. *Proceedings of the National Academy of Sciences*, *112*(27), pp.8232-8237.

Pechony, O., & Shindell, D. T. (2010). Driving forces of global wildfires over the past millennium and the forthcoming century. *Proceedings of the National Academy of Sciences*, 107(45), 19167-19170.

Portenga, E. W., Rood, D. H., Bishop, P., & Bierman, P. R. (2016). A late Holocene onset of Aboriginal burning in southeastern Australia. *Geology*, *44*(2), 131–134.

Williams, A. N., Ulm, S., Cook, A. R., Langley, M. C., & Collard, M. (2013). Human refugia in Australia during the Last Glacial Maximum and terminal Pleistocene: A geospatial analysis of the 25–12 ka Australian archaeological record. *Journal of Archaeological Science*, *40*(12), 4612–4625.

---

## Author Response (AR2)

We would like to thank Petr Kuneš and Biogeosciences for accepting our manuscript for publication as well as all reviewers for their time and contributions which allowed us toto improve the quality of the manuscript. Corrections suggested by reviewer report #1 have now been made:

- In the manuscript the notation of $CO_2$ has been corrected (subscript added)
- All figures have been reworked to ensure bigger fonts and enlargement of panels to make the maps/boxplots more legible. We hope that the quality of the figures is now of sufficient quality.

In addition, a small number of typos have been corrected in the manuscript and the supplementary information:

Line 78: *Haas et al (2002)* has been corrected to *Haas et al (2022)*

Line 279: *LGM 190 ppm* has been corrected to *LGM 185 ppm*

Supplement: *LGM 190 ppm* has been corrected to *LGM 185 ppm*